# A Century of Swine Influenza: Is It Really Just about the Pigs?

**DOI:** 10.3390/vetsci7040189

**Published:** 2020-11-26

**Authors:** Alba Frias-De-Diego, Rachael Posey, Brittany M. Pecoraro, Rafaella Fernandes Carnevale, Alayna Beaty, Elisa Crisci

**Affiliations:** 1Department of Population Health and Pathobiology, College of Veterinary Medicine, North Carolina State University, Raleigh, NC 27607, USA; afriasd@ncsu.edu (A.F.-D.-D.); bmpecora@ncsu.edu (B.M.P.); 2William Rand Kenan, Jr. Library of Veterinary Medicine, North Carolina State University, Raleigh, NC 27607, USA; rachael.posey@venebio.com; 3Department of Nutrition and Animal Production, Universidade de São Paulo, Pirassununga 13635-900, State of São Paulo, Brazil; rafaella.carnevale@usp.br; 4College of Agriculture and Life Sciences, North Carolina State University, Raleigh, NC 27695, USA; afbeaty2@ncsu.edu

**Keywords:** bibliometrics, swine influenza virus, influenza variants, GenBank, global health

## Abstract

Influenza viruses (IV) are one of the major threats to human and animal health worldwide due to the variety of species they affect. Pigs play an important role in IV ecology as the “mixing vessel,” since they can be infected by swine, avian and human IV, allowing the appearance of new subtypes. Human viruses originated in swine are known as IV of swine origin or swine influenza virus (SwIV) variants. In this study, we identified knowledge tendencies of SwIV and assessed potential bias in the literature caused by these variants. We identified the most mentioned SwIV variants and manually reviewed the literature to determine the number of publications applying the whole influenza nomenclature, a partial nomenclature, only the subtype or mixed terminology, along with the proportion of articles in which the GenBank ID number was available. We observed that the 2009 H1N1 human pandemic created an important bias in SwIV research driven by an increase in human publications on the IV of swine origin. H1N1 is the most studied subtype for swine and humans, followed by H3N2. We found differences between the nomenclatures applied, where partial classifications were slightly more common. Finally, from all the publications, only 25% stated the GenBank ID of the sequence studied. This review represents the most complete exploration of trends in SwIV knowledge to date and will serve as a guidance for future search strategies in SwIV research.

## 1. Introduction

Influenza viruses (IVs) have an undeniable impact on different species’ populations due to their wide variety of susceptible hosts and the worldwide economic impact these viruses can have. So far, IVs have only been isolated in birds and mammals, in particular influenza A viruses, in which pigs (*Sus scrofa*) have played an especially important role, being considered the “mixing vessel” of these viruses [1]. The well-known drift and shift abilities of IVs have provided them with the capacity to genetically reassort (swap genes) and cross host species’ barriers [2], which has been observed especially between human, bird and swine [3,4,5,6,7,8,9,10]. For that reason, viruses originating in swine that are able to infect other hosts are known as swine influenza virus (SwIV) variants or IVs from swine origin.

Due to the rapid changes and appearance of new IVs, their nomenclature and classification has been altered numerous times, and it was not until 1980 that a complete nomenclature composed of virus type/host species/site of origin/strain designation/year of origin and (HA -H- and NA -N-) subtype was determined by convention [11,12]. This variation in terminology seen throughout the literature, along with the terms “SwIV variant” and “IV from swine origin” has shaped the trends in SwIV research towards certain IV subtypes that do not necessarily relate to SwIV directly, but to general influenza viruses affecting hosts of particular interest due to their impact on health and economics, such as humans or production animals [13].

The authors chose to examine the literature related to swine influenza via a literature review and bibliometric analysis for a number of reasons. Literature reviews are a popular approach for knowledge synthesis in a wide variety of fields, and are commonly used to determine conceptual limits of research topics [14,15], becoming one of the key research tools in the current era of massive data production and analysis. They are especially useful when there are many studies on a topic with heterogeneous designs and results [16], and therefore they can be approached from multiple angles, making them an accessible and useful tool that can be easily achieved by the combination of several methodologies.

Bibliometric analyses have been widely applied within literature reviews to evaluate publication trends and growth in a variety of fields from social to biomedical sciences [17,18,19,20,21]. High-impact publications in the field of infectious diseases, such as the study performed by VanderWaal and Deen [22] describing the publication trends of the most relevant swine pathogens worldwide, show the applicability of these methods to improve the quality of research in the field and spot current gaps or research needs within specific topics. Another example of this approach is the study done by Nafade et al. [23], which describes the publication trends of tuberculosis (TB) by combining bibliometric analyses, manual screening and statistic tests, giving a detailed description of the global variation in the interests of TB over time. Additional studies have also applied bibliometric analyses to other questions in infectious diseases, as in the case of Wiethoelter et al. [24], where the focus was not the specific pathogen but the interaction between wildlife and livestock to characterize the most common animal species and regions involved in the wildlife-livestock interface. These approaches are based on a bibliography search followed by the automated and/or manual extraction of the information of interest from each publication. Data are computationally analyzed to obtain the evolution of the variables that shape the different research trends in the original field of interest and different software tools have been developed for conducting bibliometric analysis in science [25].

Interestingly, our searches located only two bibliometric studies of swine influenza available in the literature to date [26,27]. In contrast to the publications by Baskaran et al. [26] and Ramakrishnan et al. [27], our study applies cutting edge bibliometric research methodologies along with a more restrictive search criteria, the inclusion of research conducted in any location, worldwide and the accountability of genetic information availability. This combination allows analyses of citations to assess the evolution of research trends and interests, detect publication patterns over time worldwide, identify shifts in the focus of a given area, determine the most productive countries and their collaboration network through the analysis of large datasets [28] and identify the most common genes or genomes applied in this area of research, which has not yet been performed for SwIV. The new computational advances and informatic approaches applied to improve bibliometric research now allow the extraction and analysis of more refined and detailed information, promoting the application of bibliometrics from a great number of perspectives to identify and decrease current information gaps in different research fields.

Our study aims to evaluate and describe publication trends of “swine influenza” from 1930 to 2020 through an assessment of the available biomedical literature, assuming that publication patterns are closely related with the fluctuations in research priorities. First, we aim to identify the publication tendencies of the research conducted on swine influenza and compare it with the patterns followed by the main described hosts of SwIV to assess potential host-based bias. Then, we intend to determine the most common virus subtypes mentioned in the literature, assess the percentage of the publications that completely follow the whole influenza nomenclature designated by convention [11,12,29], and measure the proportion of those articles in which information about the genetic data used is available.

## 2. Materials and Methods

We searched SCOPUS in March 2020 for all available swine influenza publications and then performed a series of subsequent independent searches to restrain the publications to SwIV in pigs, humans, mice and ferrets, since mice and ferrets are commonly used as laboratory models for influenza studies. We also performed a second search to obtain the total number of publications of each individual IV subtype (H4N6, H1N7, H2N3, H3N2, H3N1, H1N2, H1N1), where we included influenza subtypes with avian origin that have been isolated in swine. Details of search criteria can be found in Appendix A. These searches were designed to detect publications containing our search criteria in the title, abstract or keywords. We compared the results of our search criteria between SCOPUS, Web of Science, and PubMed; from Web of Science and PubMed we obtained a more limited number of detected publications (1048 and 1032, respectively) and thus, subsequent analyses were performed using the SCOPUS output.

From each publication identified by the literature searches, we extracted the journal, year of publication, title, abstract, author names and author affiliations in .bib format. Publications for which more than one country was reported were assigned to all countries mentioned [22,23,24]. Using the R package Bibliometrix (R Foundation for Statistical Computing, Vienna, Austria) [28], we calculated the annual scientific production, country scientific production, collaboration networks, and most frequent keywords used for SwIV and the three analyzed variants. 

Finally, we performed a manual screening of the publications gathered in the most studied influenza subtypes (H1N1, H1N2, H3N2) within our SwIV downloaded articles to determine which type of nomenclature was more often applied, where the possibilities were: (i) only subtype, (ii) whole nomenclature (virus type/host species/site of origin/strain designation/year of origin and (HA -H- and NA -N- subtype)), (iii) partial nomenclature, where only part of the information of the proposed full name is available or (iv) multiple, where different forms of nomenclature were used throughout the paper. Likewise, we identified the GenBank ID numbers found in those publications and calculated the proportion of the total articles analyzed that provided this information. The data cleaning performed during this screening included the removal of duplicates and papers that could not be accessed through their listed DOI, as well as any publications in a different language than English. Finally, any lacking metadata missing for further analysis was manually added during this step.

## 3. Results

### 3.1. Publications Related to Host

#### 3.1.1. Annual Scientific Production

These analyses showed a clear peak in the number of publications of the four groups starting in 2009 that corresponds to the human H1N1 influenza pandemic. As expected, the scientific effort for all four groups was highly shaped with that pandemic, especially for swine and human viruses (Figure 1). It is also interesting to note how research on swine influenza started to rise in the 1970s, coinciding with the 1968 Hong Kong H3N2 outbreak [30], with a maximum of 16 publications per year detected from 1930 to 1969 and a total of 190 papers available for this time period.

Unsurprisingly, we found USA to be the most productive country in the analyzed timeframe (1930–2020) (1085 publications), and the country of affiliation for the author with the highest number of publications (Vincent A.L., with 63 publications detected), followed by China (310 publications, Shortridge K.F., with 16 publications). If we consider the European Union as a unit, we detected 1268 published articles, with Germany in the lead (303 articles, Dürrwald, R. with 24 publications) followed by France (226, Simon, G. with 27 publications), United Kingdom (176, Brown, I.H. with 25 publications), Spain (151, Segales, J. with 16 publications), Belgium (123, Van Reeth, K. with 40 publications) and Italy (117, Foni, E. with 16 publications) (Appendix A). However, these values indicate the amount of publications in which at least one author’s affiliation corresponds to a given country. For that reason, when countries are grouped as in the case of the European Union, papers with multiple European affiliation will be accounted more than once.

#### 3.1.2. Collaboration Networks

International network analysis showed the USA as the country with broadest collaboration network, presenting scientific relationships with 72 other countries. United Kingdom and China had the second and third highest number of collaborations (56 and 41 countries) followed by Canada and Spain (39 countries) (Figure 2, Appendix A).

#### 3.1.3. Keywords

The keyword set detected by our analyses was similar between all groups, with the most common keywords found including influenza A virus, swine and animals (Figure 3).

### 3.2. Publications Related to SwIV Subtypes

#### 3.2.1. Most Common Subtypes

We found that 89% of the total number of studies detected in pigs (1253) related to H1N1, 9.8% related to H3N2, 1% to H1N2, 0.2% to H3N1, and no publications were detected for the other subtypes (Figure 4). The manual screening of these publications showed that only 11.8% of them applied the entire influenza nomenclature virus type/host/geographic origin/strain number/year of isolation/(virus subtype), while 28.2% used the virus subtype and 34.4% presented a partial nomenclature missing one or more sections of the name. In addition, 16% of the screened studies applied more than one different nomenclature through the paper and 24.4% were either in a non-English language or non-accessible through the obtained DOI (Figure 5).

#### 3.2.2. Available Genetic Information

The manual screening for this section was performed as a subsequent selection of the database analyzed on the previous section. For this screening, we analyzed all publications with a valid DOI, leaving the initial screening with a total of 1108 results. We found that only 25% of these papers provided genetic information of the studied virus via disclosure of GeneBank IDs (see Appendix A for DOIs and corresponding accession numbers).

## 4. Discussion

Influenza virus (IV) has been one of the major and most recurrent targets of biomedical and veterinary science over time, especially after the 2009 pandemic, which triggered an interest in and growth in published literature on swine influenza viruses (SwIV) and on viruses originating in swine that are able to infect other species (known as IV of swine origin or SwIV variants). This interest growth in SwIV has promoted the origin of multiple scientific networks and has influenced the number of publications for SwIV and its variants (as seen in Figure 1). 

There is an undeniable weight on research efforts targeted to species that are somehow influential to humankind, either from a health or an economic point of view [31], which raises the question of whether research on a certain topic may be biased toward those species. For that reason, a literature review along with bibliometric analyses showing this variability worldwide will help detect and understand these biases in a given area of research. Surprisingly given the impact of this virus worldwide, we only found two previous partial explorations of SwIV research [26,27]. However, more restrictive search criteria and a worldwide assessment of a century of research make this literature review the most detailed assessment of SwIV to date.

Bibliometric approaches are increasingly becoming more common for biomedical sciences [17,18,19,20,21], including veterinary medicine, where articles such as the ones published by VanderWaal and Deen [22] or Wiethoelter et al. [24] describing research trends of swine pathogens worldwide and its relationship with wildlife, are exhibiting the applicability of bibliometric studies as an efficient screening method to detect a wide variety of gaps and research needs.

The bibliographic search was performed using the SCOPUS database, as it is a large, multidisciplinary database that includes MEDLINE and focuses on natural sciences and biomedical research [32]. As expected, the number of publications focusing on each group (swine, human, mouse and ferret) decreased as the criteria increased its restrictiveness (Figure 1). Even though we did not restrict our search to a single influenza type (A, B or C), the obtained publications had a clear focus on Influenza A viruses. We observed that although each group presented an individual publication pattern, they were all highly influenced by the 2009 pandemic (Figure 1). Interestingly, the first observed peak in scientific production corresponds to the period following the H3N2 Hong Kong pandemic in 1968 [30], showing that, indeed, disease events affecting humankind correlate to sharp increases in research efforts not only in humans, but also in the most common species used for research within the subtype causing the outbreak. Based on these results, further bibliometric research could focus on the identification of specific research interest of individual subtypes overtime outside of both pandemics.

As with the annual scientific production, the general collaboration networks for all groups is biased by the international partnerships found in human research (Figure 2). Unsurprisingly, the USA was the country with the highest number of international collaborations (42) (Appendix A), which was an expected result since it is the country with highest investment in R&D worldwide [33]. Interestingly, from the 15 most productive countries shown in Appendix A, half of them are members of the European Union. These countries also shared a dense collaboration network, which is clearly visible in Figure 2. In contrast, the rapid scientific growth of emerging economies such as BRICS countries (Brazil, Russia, India and China) over time has been previously mentioned [23,34]. This predicted emergence is noticeable in our results, where we have found China and Brazil as the 2nd and 3rd countries with more international collaborations, which has a direct relationship with the number of publications per country, where China was also seen the second most productive country (Appendix A). Our analyses also confirmed the hypothesis proposed by Nafade et al. [23] that collaboration between BRICS countries were less frequent than collaborations between BRICS countries and high-income countries, helping the less developed to improve the quantity and impact of their research (Appendix A).

As seen in previous results, the occurrence of keywords was slightly different for all groups, with noticeable influence of human research shown by the presence of “human” as a keyword in all four groups. Despite the differences found, multiple words were common for all groups such as influenza A virus, swine and animals (Figure 4). This slight variability was expected, since although we are analyzing different variants, they are all part of SwIV research. However, this may also hinder the ability to locate and analyze more specific articles within the topic.

The analysis of the most common subtypes within SwIV showed H1N1 as the most studied, followed by H3N2 (Figure 4). This was an expected result since H1N1 was not only the causative agent of the 2009 pandemic but also of the Spanish flu pandemic in 1918, which is still considered one of the most severe in human history [35,36]. This subtype has also been the cause of numerous other outbreaks in different species occurring to the present day [37]. Likewise, H3N2 was the causative agent of the Hong Kong outbreak in 1968, which has been described as a reassortment between previously circulating human influenza and avian viruses that most likely happened with pigs as the “mixing vessel” [30,38]. Future analyses could address the potential shift of research interest between avian and swine influenza triggered by different pandemics over time.

Given that our first analyzed publications are dated from 1930, and given the great impact of this subtype over time, it was expected that the number of publications related to H1N1 would be remarkably high. Finally, as seen in Figure 5, we did not find that one type of nomenclature was heavily predominant, although partial or non-complete nomenclatures were slightly more prevalent since the whole nomenclature is usually only mentioned once or twice in a publication, followed by the use of the simple subtype or a shorter version. Interestingly, although Anderson et al. [39] proposed a global nomenclature system for the H1 genes of swine influenza which may have increased the number of publications applying it, it has only been available for four years, making it very unlikely to cause a difference in comparison to the 86 remaining years included in our study. This approach is not unexpected, since the use of reduced classification facilitates article reading and understanding. However, some studies do not use the complete name at all, being in some cases descriptions of field isolates and potential new strains, leaving out important pieces of information that could be applied for other types of analyses in related fields such as epidemiology or disease surveillance. Likewise, the small percentage of the analyzed publications in which the genetic information of the viruses was available, shows one of the current biggest limitations in science, where genetic information is increasingly becoming one of the pillars for research in a wide variety of areas. For that reason, here we highlight the importance of the release of detailed information for future research, such as dates and location of collected data, genetic information and accession numbers among others, and we encourage the application of the complete nomenclature to allow further analysis and application of these data.

We found a number of limitations during the development of this study. The first one relates to the fact that the affiliation information extracted by the Bibliometrix package is not necessarily the same as the country where the research was developed or conducted, especially in the case of papers involving more than one country. However, this limitation is common in bibliometric analyses, and has been previously assessed via sensitivity analyses with no significant changes on the obtained results [23]. Another limitation was related to the use of the SCOPUS database and the search criteria applied. The combination of multiple databases for the initial download of publications would have broadened the search for articles that could have been included in the paper. Likewise, our restrictive search criteria was a trade-off between the amount of publications detected and the specificity of the results in relation to swine influenza alone, matching the scope of the paper. The last limitation encountered was the unavailability of a portion of the publications detected for the manual screening due to language and incomplete or partial reports.

Finally, we present this study as a global exploration of the knowledge trends of swine Influenza, its most common variants and the most applied nomenclature, and expect that this report will improve the search strategies and descriptions of future swine influenza research.

## 5. Conclusions

This study represents the most complete exploration of research trends and patterns in SwIV to date, considering the most applied nomenclature, the most used subtypes, and the current availability of genetic information. Our results can be used as a reference to standardize the use of the applied SwIV nomenclatures and promote the release of genetic information in original articles, serving as a guidance for future search strategies in SwIV research.

## Figures and Tables

**Figure 1 vetsci-07-00189-f001:**
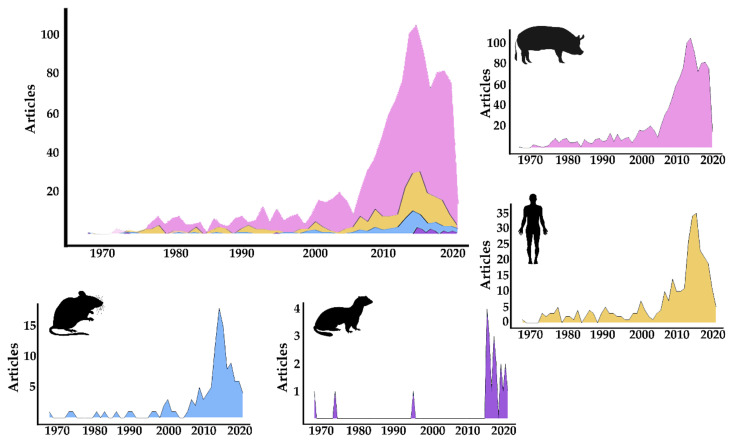
Representation of the annual scientific production in number of publications per year of SwIV research based on the host. Pink = swine; Orange = human; Blue = mice; Purple = ferrets. Top left panel represents the combination of the four hosts analyzed in this study, while each individual surrounding graph represents the particular scientific production of the publications related to the host represented through its silhouette and the above described colors.

**Figure 2 vetsci-07-00189-f002:**
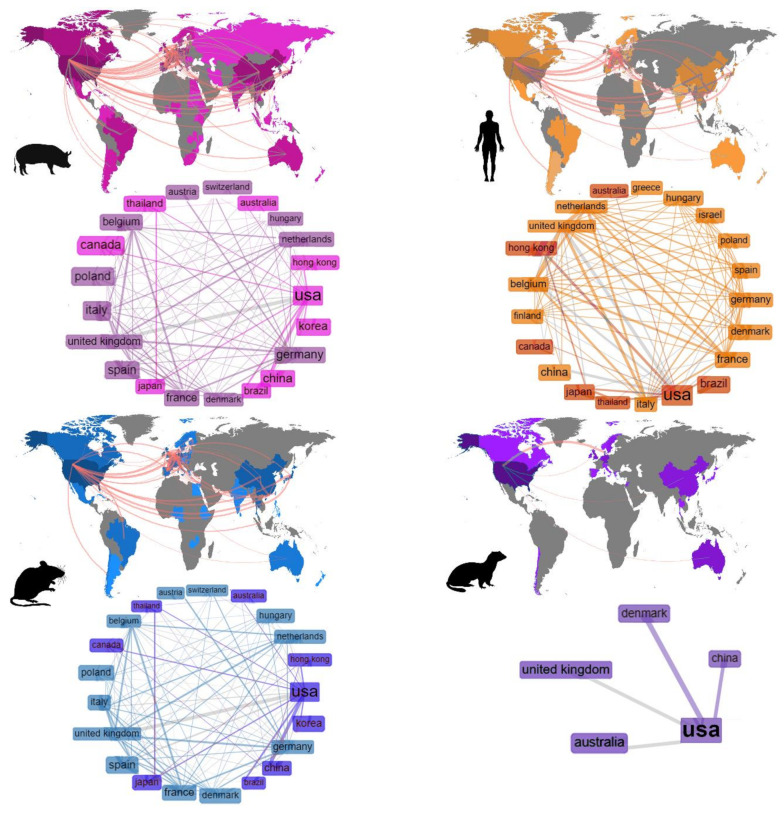
Maps showing the international collaboration relationships found in the publications of SwIV for each host group. Pink = swine; Orange = human; Blue = mice; Purple = ferrets.

**Figure 3 vetsci-07-00189-f003:**
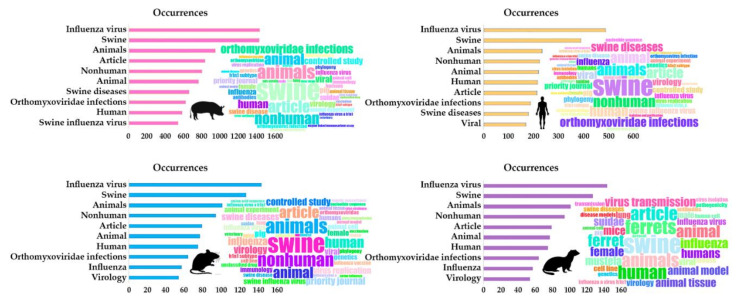
Representation of the most used keywords in the publications of each described SwIV variant based on studied hosts. Pink = swine; Orange = human; Blue = mice; Purple = ferrets.

**Figure 4 vetsci-07-00189-f004:**
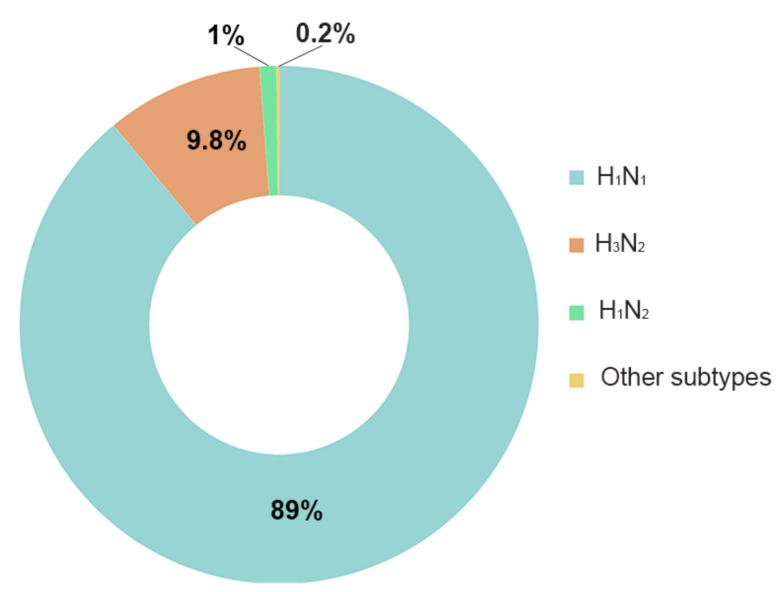
Chart showing the number of publications detected by SCOPUS linked to the different SwIV subtypes. Blue = H1N1, Orange = H3N2, Green = H1N2 and Yellow = Other analyzed subtypes.

**Figure 5 vetsci-07-00189-f005:**
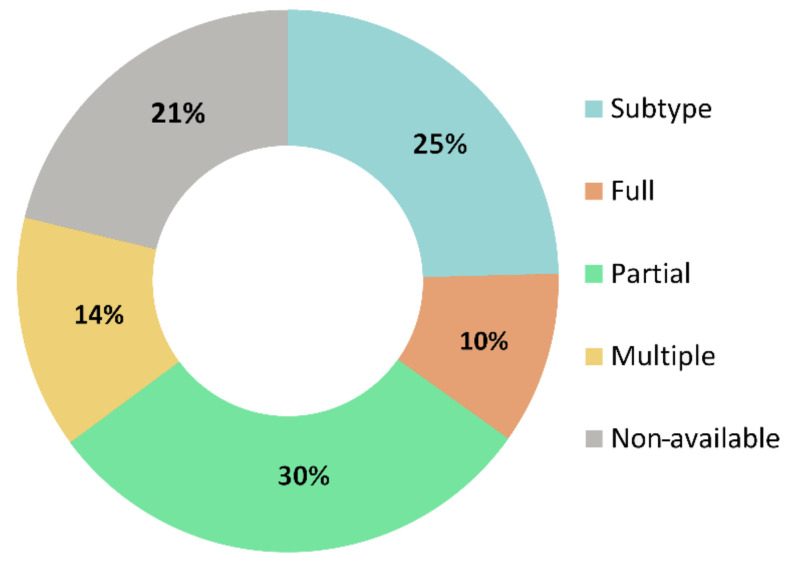
Chart representing the different nomenclatures applied on the manually screened publications. Light blue = publications only mentioning subtype, orange = publications applying full nomenclature, green = publications applying a partial nomenclature (only part of the information from the suggested name format), yellow = multiple nomenclatures (publications using more than one type of nomenclature through the paper) and grey = non-available publications.

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
