# Peer review of "A Century of Swine Influenza: Is It Really Just about the Pigs?"

_vetsci, 2020, doi:10.3390/vetsci7040189_

Round 1

Reviewer 1 Report

“A Century of Swine Influenza: Is It really just about Pigs?” by Frias-De-Diego et al., performed a retrospective bibliometric analysis of influenza A virus literature. Their main goal was to identify knowledge tendencies of SWIV and assess potential bias in literature caused by the influenza virus variant nomenclature. The authors state “their study applies cutting edge methodologies on bibliometric research” and thus allowed for the assessment of “the evolution of research trends and interest, detect publication patterns over time worldwide, identify shifts in the focus of a given area, determine the most productive countries and their collaboration network through the analysis of large datasets, and identify the most common genes or genomes applied in this area of research”. Their study aims to evaluate and describe publication trends of “swine influenza” over a 90 year span by identifying the publication tendencies of the research conducted on swine influenza and compare it to the patterns followed by the main described hosts of SwIV to assess potential host-based bias. Consequently, they aim to determine the most common virus subtypes mentioned in the literature, assess the percentage of the publications that completely follow the whole influenza nomenclature designated by convention, and measure the proportion of those articles in which information about the genetic data used is available.

Below are some of my critiques of the manuscript and suggestions for improvement.

  1. The authors report a total of 1,253 publications for swine influenza, 318 for human, 118 for mice, and 60 for ferrets between the years of 1930 to 2020. Is this a limitation of the sources available through Scopus? Or the search criteria used? Using NCBI, the search criteria ‘swine influenza’ returns 60,000 articles by itself, ‘H1N1’ returns 21,000, and ‘2009 pandemic’ returns 9,000. Either the authors’ search criteria are extremely limiting, or there is something else preventing the analysis of the real body of work publicly available.
  2. The paper presented interesting findings about swine influenza research such as that only a small percentage of papers selected in this analysis included the complete nomenclature or the GenBank ID of the sequence of the viruses studied. However it lacked novelty (it is expected that after an influenza pandemic originated from a swine variant, influenza research will be focused in viruses from the swine host that have zoonotic potential) and the analysis presented in the study appears to be swallow in relation to the extent of papers reviewed in the publication (almost 1800 studies included). Limited statistical analysis was performed and despite the fact that they reviewed numerous papers the vision of uncovering potential correlations and trends between the selected publications was narrow (e.g. examine if there is a strain selection bias).
  3. The authors identified the inclusion criteria for the articles in which they drew their conclusions. However, their criteria is never addressed as a limitation to their study. i.e the authors only included publications written in English and articles that were identified on the Scopus database which limits their sample size to a very small fraction of what is actually published. Although this is not a weakness to the article in and of itself, it is a limitation to their findings and a limitation to the application of the conclusions drawn. This type of limitation should be addressed in the body of the article.
  4. As expected, publications focused in avian influenza research were not included in the bibliometric analysis of the study. Both avian and swine origin viruses play a pivotal role in the ecology and evolution of influenza viruses. While they mention in their abstract that the study aims to “assess potential bias in literature caused by these (swine) variants” they failed to recognize that extended influenza research focused in avian viruses is ongoing. It could be actually rather interesting to assess a potential swift in the dominance of influenza research from avian to swine viruses after the 2009 pandemic.
  5. The authors manually screened the literature to determine which type of nomenclature was applied and to identify the GenBank ID numbers (when applicable). This method requires a lot of time and effort and is commendable. The questions the authors seek to understand are definitely intriguing and the data evaluated/extrapolated from their search are suitable, to some extent, to answer those questions. However, I am not convinced that the utilization and interpretation of the data truly add to the scientific community.
  6. Bibliometrix works with data extracted from four main bibliographic databases: SCOPUS, Clarivate Analytics Web of Science, Cochrane Database of Systematic Reviews, and RISmed PubMed/MedLine. I would have liked to see the authors utilize more than one database to broaden their search for applicable and acceptable articles for their study.
  7. I would like the authors to include the types of statistical analyses utilized to support the claims that there were differences and whether those findings were significant. The authors did not mention statistical analyses performed but discusses “differences in the nomenclatures applied where partial classifications were slightly more common”—are these differences statistically significant? Additionally, correlation analysis could potentially be useful in understanding the publication trends in countries with respect to the nomenclature utilized, etc. This type of analysis would positively add to the manuscripts impact of this new knowledge with respect to their goal of identifying publication tendencies and trends worldwide.
  8. The figures are extremely low resolution, such that the axis labels were difficult to read, even zoomed in. The pie charts could be reduced in size but the other graphs need to be increased a decent amount for legibility.
  9. In few of the graphs included it wasn’t mentioned clearly if the charts were about swine host or swine origin viruses.

Below are some minor comments that I believe require attention by the editor and the authors.

  1. In Section 2. Methods, the authors performed a literature search using the Scopus database. I believe that it may have been beneficial to use multiple peer-reviewed databases to perform the literature review to minimize selection bias.
  2. In Section 2. Methods, during the screening process of the literature search, the authors removed any papers that were not presented in English. I can understand why this was performed for practical reasons, however, I am concerned that interesting metadata may have been excluded from the study, solely based on language.
  3. In figure 1, the author presents a “representation of the annual scientific production in number of publication per year”, it would be beneficial for reader to label each of the area charts A, B, C, D etc.. and explain what species is being presented. This is should also be repeated in Figures 2 and 3.
  4. Figures 1, 2 and 3 in the manuscript also need to be uploaded with a higher definition, it is difficult to read the text on the graphs.
  5. In Figure 5 – it should be edited to “Non-available” on the ring chart
  6. Line 36, the authors use the term Influenza viruses which is very general. This paper is clearly about Influenza A viruses and this distinction is important. The authors should correct their terminology throughout to reflect this.
  7. Line 99 the authors mention the subtypes included in their search as H4N6, H1N7, H2N3, H3N2, H3N1, H1N2 and H1N1. Of these, H4N6, H1N7 and H2N3 are avian viruses, not swine. The swine viruses are the remaining H1N1, H1N2, H3N2, and H3N1 subtypes. The authors later narrow their criteria to only include H1N1, H1N2, and H3N2. Either remove the erroneous subtypes or more clearly define why these subtypes were picked. The avian species mentioned have been isolated from swine, so they could be included in the scope of this paper, but must be listed as such.

Author Response

Reviewer 1

Below are some of my critiques of the manuscript and suggestions for improvement.

1. The authors report a total of 1,253 publications for swine influenza, 318 for human, 118 for mice, and 60 for ferrets between the years of 1930 to 2020. Is this a limitation of the sources available through Scopus? Or the search criteria used? Using NCBI, the search criteria ‘swine influenza’ returns 60,000 articles by itself, ‘H1N1’ returns 21,000, and ‘2009 pandemic’ returns 9,000. Either the authors’ search criteria are extremely limiting, or there is something else preventing the analysis of the real body of work publicly available.

We understand the concern of the reviewer about this topic. Along this process, we tried different search criteria with different levels of stringency. Searching publications only under the term “swine influenza” detected many publications in which this disease was mentioned in the abstract as an example of either disease or virus, even though the actual publication topic was a different one. For that reason, we needed to create a search criteria that restricted the amount of publications found to ensure that each of those were part of swine influenza research, and not only related to the word influenza or the subtypes themselves. Since those words are general, they are easily found in abstract, keywords or even titles, even if the article doesn’t study them specifically.

However, we have mentioned this in the limitations of the study, in Lines 281-286:

“Another limitation was related to the use of the SCOPUS database and the search criteria applied. The combination of multiple databases for the initial download of publications would have broadened the search for articles that could have been included in the paper. Likewise, our restrictive search criteria was a trade-off between the amount of publications detected and the specificity of the results in relation to swine influenza alone, matching the scope of the paper.”

2. The paper presented interesting findings about swine influenza research such as that only a small percentage of papers selected in this analysis included the complete nomenclature or the GenBank ID of the sequence of the viruses studied. However it lacked novelty (it is expected that after an influenza pandemic originated from a swine variant, influenza research will be focused in viruses from the swine host that have zoonotic potential) and the analysis presented in the study appears to be swallow in relation to the extent of papers reviewed in the publication (almost 1800 studies included). Limited statistical analysis was performed and despite the fact that they reviewed numerous papers the vision of uncovering potential correlations and trends between the selected publications was narrow (e.g. examine if there is a strain selection bias).

We thank the reviewer for the comment. We consider that our publication’s novelty relies on the general bibliometric study of the research trends of swine influenza rather than in the expectable influence of the pandemic in human research. Interestingly, research of flu specifically in pigs was not extensive after the pandemic of 1918, opposed to the 2009 pandemic.

Furthermore, the small percentage of GenBank ID numbers found is related to the fact that the even though GenBank appeared in 1982 (NIH, 2008), IDs started to be more often reported in the last decades. The first article presenting a GenBank ID in our database was 1991, then one on 1995 and another one in 1997.

Statistical concern of the reviewer was addressed in comment 7.

National Institutes of Health (2008). GenBank Celebrates 25 Years of Service with Two-Day Conference; Leading Scientists Will Discuss the DNA Database at April 7-8 Meeting. Online Access via this link.

3. The authors identified the inclusion criteria for the articles in which they drew their conclusions. However, their criteria is never addressed as a limitation to their study. i.e the authors only included publications written in English and articles that were identified on the Scopus database which limits their sample size to a very small fraction of what is actually published. Although this is not a weakness to the article in and of itself, it is a limitation to their findings and a limitation to the application of the conclusions drawn. This type of limitation should be addressed in the body of the article.

We agree with the reviewer and have addressed the suggestion by adding the following text in Lines 281-286 (also mentioned in Reviewer’s comment 1):

Another limitation was related to the use of the SCOPUS database and the search criteria applied. The combination of multiple databases for the initial download of publications would have broadened the search for articles that could have been included in the paper. Likewise, our restrictive search criteria was a trade-off between the amount of publications detected and the specificity of the results in relation to swine influenza alone, matching the scope of the paper. The last limitation encountered was the unavailability of a portion of the publications detected for the manual screening due to language and incomplete or partial reports.”

4. As expected, publications focused in avian influenza research were not included in the bibliometric analysis of the study. Both avian and swine origin viruses play a pivotal role in the ecology and evolution of influenza viruses. While they mention in their abstract that the study aims to “assess potential bias in literature caused by these (swine) variants” they failed to recognize that extended influenza research focused in avian viruses is ongoing. It could be actually rather interesting to assess a potential swift in the dominance of influenza research from avian to swine viruses after the 2009 pandemic.

We agree with the reviewer's comment. In fact, in our previous version of the paper we did consider avian influenza (we are attaching the obtained figures below: Figure 1,2,3). (See attached file)

We wanted to recognize the extension of avian viruses, however, even though swine can be affected by avian influenza viruses, these results were not matching the swine influenza scope of the publication. Avian viruses were creating a bias in the literature review and this was also one of the reasons why we decided to restrict even more our research criteria, to make sure that every publication detected was related to swine influenza specifically and not including also avian influenza. We consider that the avian viruses data should not be included in the current paper and considering the reviewer’s comment we have also added the following text in L 254-255:

Future analyses could address the potential swift of research interest between avian and swine influenza triggered by different pandemics over time.”

Figure 1: Annual scientific publications found for papers on avian influenza from 1930 to 2020.

Figure 2: Collaboration network of the detected publications on avian influenza from 1930 to 2020.

Figure 3: Most commonly used keywords on the detected publications on avian influenza from 1930 to 2020.

5. The authors manually screened the literature to determine which type of nomenclature was applied and to identify the GenBank ID numbers (when applicable). This method requires a lot of time and effort and is commendable. The questions the authors seek to understand are definitely intriguing and the data evaluated/extrapolated from their search are suitable, to some extent, to answer those questions. However, I am not convinced that the utilization and interpretation of the data truly add to the scientific community.

This section of our manuscript aims to demonstrate the lack of genetic information available for a relevant pathogen for both human and animal health, even though science is rapidly moving towards the application of genetic data for medical and epidemiological applications.

6. Bibliometrix works with data extracted from four main bibliographic databases: SCOPUS, Clarivate Analytics Web of Science, Cochrane Database of Systematic Reviews, and RISmed PubMed/MedLine. I would have liked to see the authors utilize more than one database to broaden their search for applicable and acceptable articles for their study.

We understand the reviewer’s opinion. We chose SCOPUS as our working database since a variety of studies have described it as the database with more coverage in the biomedical field, which led us to think that most of the publications detected about a pathogen like swine influenza would overlap between databases, adding a small percentage of new publications to our search.

For example, when we performed the same search in Web of Science, we only obtained 1,048 publications, which was a more limited search outcome than the one obtained through SCOPUS. This also happened for PubMed, from which we could only retrieve 1,032 publications.

For that reason, we have added the following text in the manuscript (lines 105-108)

We compared the results of our search criteria between SCOPUS, Web of Science, and PubMed, from which we obtained a more limited number of detected publications (1,048 and 1,032 respectively) and thus, subsequent analyses were performed using the SCOPUS output.”

We have also mentioned this in the limitations of our study (Lines 282-284):
The combination of multiple databases for the initial download of publications would have broadened the search for articles that could have been included in the paper”

7. I would like the authors to include the types of statistical analyses utilized to support the claims that there were differences and whether those findings were significant. The authors did not mention statistical analyses performed but discusses “differences in the nomenclatures applied where partial classifications were slightly more common”—are these differences statistically significant? Additionally, correlation analysis could potentially be useful in understanding the publication trends in countries with respect to the nomenclature utilized, etc. This type of analysis would positively add to the manuscripts impact of this new knowledge with respect to their goal of identifying publication tendencies and trends worldwide.

We thank the reviewer for the comment and understand the relevance of statistical analysis for original articles. However, our manual screening only provided one value per group of publications, meaning we only have one single numerical value for whole nomenclature (147 papers counted), one single value for subtype only (351 papers) and so on.

For example, we have that:

  • Subtype = 351
  • Full nomenclature = 147
  • Partial nomenclature = 428

Since our whole database for statistic analyses would be n=3, where n1 = 351, n2=147 and n3 = 428, there are no tests we can perform that give any sort of significance, since even to do descriptive statistics we would need to have replicates of the same group to calculate means, standard deviations etc.

For that reason, and to avoid confusion, we used the percentage of the total of articles detected and avoided to use the term “significant” to refer to any of the differences found in our screenings.

8. The figures are extremely low resolution, such that the axis labels were difficult to read, even zoomed in. The pie charts could be reduced in size but the other graphs need to be increased a decent amount for legibility.

Figures were changed accordingly.

9. In few of the graphs included it wasn’t mentioned clearly if the charts were about swine host or swine origin viruses.

We have addressed this by modifying the figure descriptions.

Below are some minor comments that I believe require attention by the editor and the authors.

1. In Section 2. Methods, the authors performed a literature search using the Scopus database. I believe that it may have been beneficial to use multiple peer-reviewed databases to perform the literature review to minimize selection bias.

We agree with the reviewer’s comment and have addressed that concern in our limitations section (Lines 281-286)

2. In Section 2. Methods, during the screening process of the literature search, the authors removed any papers that were not presented in English. I can understand why this was performed for practical reasons, however, I am concerned that interesting metadata may have been excluded from the study, solely based on language.

We understand the reviewer’s concern.

English is currently the universal language applied for scientific work. Furthermore, trying to translate a foreign language without the proper knowledge would lead to misinterpretation of the data published, particularly in a publication like ours in which we need to manually extract the available metadata, creating unnecessary bias and potential mistakes interpreting the data context.

3. In figure 1, the author presents a “representation of the annual scientific production in number of publication per year”, it would be beneficial for reader to label each of the area charts A, B, C, D etc.. and explain what species is being presented. This is should also be repeated in Figures 2 and 3.

We have addressed this by modifying the figure description.

4. Figures 1, 2 and 3 in the manuscript also need to be uploaded with a higher definition, it is difficult to read the text on the graphs.

Figures were changed accordingly.

5. In Figure 5 – it should be edited to “Non-available” on the ring chart

The figure was changed accordingly.

6. Line 36, the authors use the term Influenza viruses which is very general. This paper is clearly about Influenza A viruses and this distinction is important. The authors should correct their terminology throughout to reflect this.

We agree with the reviewer and have added the following text into our manuscript introduction (Lines 36-38):

So far, IVs have only been isolated in birds and mammals, in particular Influenza A viruses, from which pigs (Sus scrofa) have played an especially important role”

And in the discussion section (Lines 216-218).

Interestingly, even though we did not restrict our search to a single influenza type (A, B or C), the obtained publications had a clear focus on Influenza A”

7. Line 99 the authors mention the subtypes included in their search as H4N6, H1N7, H2N3, H3N2, H3N1, H1N2 and H1N1. Of these, H4N6, H1N7 and H2N3 are avian viruses, not swine. The swine viruses are the remaining H1N1, H1N2, H3N2, and H3N1 subtypes. The authors later narrow their criteria to only include H1N1, H1N2, and H3N2. Either remove the erroneous subtypes or more clearly define why these subtypes were picked. The avian species mentioned have been isolated from swine, so they could be included in the scope of this paper, but must be listed as such.

H4N6, H1N7 and H2N3 are viruses that were isolated in swine, as described in the references below. That is the reason why they were included in our analyses.  The restriction of the criteria done in the paper is not based on the origin of these viruses, but on the influenza subtypes isolated in swine and the number of publications found for them. That is the reason why the publications on H1N1, H1N2, and H3N2 were applied for subsequent analyses.

Following the suggestion of the reviewer, to clarify this aspect, we have added the following text in lines 101-103:

We also performed a second search to obtain the total number of publications of each individual IV subtype (H4N6, H1N7, H2N3, H3N2, H3N1, H1N2, H1N1), where we included influenza subtypes with avian origin that have been isolated in swine.”

H4N6  Karasin, A.I., Brown, I.H., Carman, S. and Olsen, C.W., 2000. Isolation and characterization of H4N6 avian influenza viruses from pigs with pneumonia in Canada. Journal of virology, 74(19), pp.9322-9327.

H1N7 Brown, I.H., Alexander, D.J., Chakraverty, P., Harris, P.A. and Manvell, R.J., 1994. Isolation of an influenza A virus of unusual subtype (H1N7) from pigs in England, and the subsequent experimental transmission from pig to pig. Veterinary microbiology, 39(1-2), pp.125-134.

H2N3  Ma, W., Vincent, A.L., Gramer, M.R., Brockwell, C.B., Lager, K.M., Janke, B.H., Gauger, P.C., Patnayak, D.P., Webby, R.J. and Richt, J.A., 2007. Identification of H2N3 influenza A viruses from swine in the United States. Proceedings of the National Academy of Sciences, 104(52), pp.20949-20954.

Reviewer 2 Report

The manuscript presented by Frias-De-Diego et al. describes a bibliometric analysis of swine influenza publications from 1930-2020. In this study the authors identified trends in the host species, and swine influenza subtypes studied during this time period, as well as the geographical distribution of the publications and associated author collaborations. The authors also describe how the viral nomenclature is used within the publications and how frequently  viral genetic sequences are made available. Overall, the study highlights interesting historical trends regarding swine influenza publications.

Minor Comments

Throughout - the authors should clarify that ferrets and mice are commonly used as laboratory models for studying influenza viruses, rather than host species for the isolation of novel/circulating influenza viruses.

Throughout - the authors have used the term serotype to refer to the influenza virus type and subtypes in the manuscript, which is incorrect. The authors should replace serotype with the correct term (type or subtype) where appropriate.

Lines 60 - 84 - the authors should move this section to the discussion, as it is comparing the current study with other in the literature, rather than outlining the current study. Replace with an overview of bibliometric methods and analyses to introduce the reader to this topic.

Line 130 - the authors should clarify the time period indicated and the end of this line, as it is not clear.

Line 132-133 - the authors should clarify how the different graphs/colours relate to the difference species identified. Does this figure relate to the species studied in the publications or the species from which the viruses were isolated?

Line 154 - remove the word 'fairly' as this is too ambiguous.

Lines 160 - 167 - the authors have looked at the use of nomenclature in the publications but were there any differences observed over the time period studied? It would be beneficial for the reader to illustrate if there were any trends in nomenclature used throughout this time period. I know this is partially address in the introduction (lines 44-51), but it would be of interest to understand any quantitative differences in the use of nomenclature. Also, Anderson et al. (PMID:27981236) published a global nomenclature system for the H1 genes of swine influenza viruses in 2016. Do the authors observe any trends with regards to the use of nomenclature around the publishing of similar suggested nomenclature systems?

Lines 160-167 - The authors present the distribution of publications focused on particular subtypes, but do the authors observe any variations in the subtypes studied/identified with relation to time as well? This is partially answered when discussing the 2009 H1N1 pandemic and H3N2 Hong Kong pandemic, but are there any other trends?

Author Response

Reviewer 2

Minor Comments

Throughout - the authors should clarify that ferrets and mice are commonly used as laboratory models for studying influenza viruses, rather than host species for the isolation of novel/circulating influenza viruses.

We agreed with the reviewer comment and have added the following text in lines 99-101:

“...searches to restrain the publications to SwIV in pigs, humans, mice and ferrets, since mice and ferrets are commonly used as laboratory models for the study of influenza“

Throughout - the authors have used the term serotype to refer to the influenza virus type and subtypes in the manuscript, which is incorrect. The authors should replace serotype with the correct term (type or subtype) where appropriate.

We agree with the reviewer and have corrected the text substituting the term serotype by subtype through the whole manuscript.

Lines 60 - 84 - the authors should move this section to the discussion, as it is comparing the current study with other in the literature, rather than outlining the current study. Replace with an overview of bibliometric methods and analyses to introduce the reader to this topic.

We thank the reviewer for the comment. We consider that this specific section of the introduction is needed to provide the appropriate context to understand the rest of the manuscript and consider that it would be detrimental to move it all to the discussion.

However, we agree with the reviewer in the fact that the introduction needed an overview of the bibliometric methods, and therefore we added the following text in lines 71-75:

“These approaches are based on a bibliography search followed by the automated and/or manual extraction of the information of interest from each publication. Data are computationally analyzed to obtain the evolution of the variables that shape the different research trends in the original field of interest and different software tools have been developed for conducting bibliometric analysis in science [25]

We also agree with the reviewer in the fact that the discussion needed to have a summary of what was stated in the introduction, and for that reason we have added the following text in lines 208-212:

Bibliometric approaches are increasingly becoming more common for biomedical sciences [17-21], including veterinary medicine, where articles such as the ones published by VanderWaal and Deen [22] or Wiethoelter et al., [24] describing research trends of swine pathogens worldwide and its relationship with wildlife are exhibiting the applicability of bibliometric studies as an efficient screening method to detect a wide variety of gaps and research needs.”

Line 130 - the authors should clarify the time period indicated and the end of this line, as it is not clear.

We changed the manuscript accordingly.

Line 132-133 - the authors should clarify how the different graphs/colours relate to the difference species identified. Does this figure relate to the species studied in the publications or the species from which the viruses were isolated.

We have addressed this by modifying the figure descriptions.

However, since the R package extracts the information from Title, Abstract, and keywords, publications describing a particular host for the virus as well as the species from which one virus has been isolated will be equally detected (as long as the terms written in our search criteria can be found in either Title, Abstract, or keywords).

Line 154 - remove the word 'fairly' as this is too ambiguous.

The text was changed accordingly.

Lines 160 - 167 - the authors have looked at the use of nomenclature in the publications but were there any differences observed over the time period studied? It would be beneficial for the reader to illustrate if there were any trends in nomenclature used throughout this time period. I know this is partially address in the introduction (lines 44-51), but it would be of interest to understand any quantitative differences in the use of nomenclature. Also, Anderson et al. (PMID:27981236) published a global nomenclature system for the H1 genes of swine influenza viruses in 2016. Do the authors observe any trends with regards to the use of nomenclature around the publishing of similar suggested nomenclature systems?

We found interesting the reviewer’s comment and checked it in our database.

We found that 99 papers applied the whole nomenclature from 1930-2016, while only 47 publications used this nomenclature between 2016-2020.
Although the system proposed for H1 genes may have increased the amount of authors deciding to use this nomenclature, given that it has only been available for four years it was very unlikely to have already caused a difference in comparison to the 86 remaining years that were included in our study.

We have added the findings in the 261-264 of the manuscript.

Interestingly, although Anderson et al., (2016) proposed a global nomenclature system for the H1 genes of swine influenza which may have increased the number of publications applying it, it has only been available for four years, making it very unlikely to cause a difference in comparison to the 86 remaining years included in our study”

Lines 160-167 - The authors present the distribution of publications focused on particular subtypes, but do the authors observe any variations in the subtypes studied/identified with relation to time as well? This is partially answered when discussing the 2009 H1N1 pandemic and H3N2 Hong Kong pandemic, but are there any other trends?

We agree with the reviewer’s comment and even though we have not analyzed this in detail due to time constraints, we have added the following text in the discussion (line 223-225):

Based on these results, further bibliometric research could focus on the identification of specific research interest of individual subtypes overtime outside of both pandemics”